# The epidemiology of extrapulmonary tuberculosis in China: A large-scale multi-center observational study

Wanli Kang[1☯], Jiajia Yu[1☯], Jian Du[1☯], Song Yang[2☯], Hongyan Chen[3], Jianxiong Liu[4], Jinshan Ma[5], Mingwu Li[6], Jingmin Qin[7], Wei Shu[1], Peilan Zong[8], Yi Zhang[9], Yongkang Dong[10], Zhiyi Yang[11], Zaoxian Mei[12], Qunyi Deng[13], Pu Wang[14], Wenge Han[15], Meiying Wu[16], Ling Chen[17], Xinguo Zhao[18], Lei Tan[19], Fujian Li[20], Chao Zheng[21], Hongwei Liu[3], Xinjie Li[4], Ertai A[5], Yingrong Du[6], Fenglin Liu[7], Wenyu Cui[9], Quanhong Wang[10], Xiaohong Chen[11], Junfeng Han[12], Qingyao Xie[13], Yanmei Feng[14], Wenyu Liu[15], Peijun Tang[16], Jianyong Zhang[17], Jian Zheng[18], Dawei Chen[20], Xiangyang Yao[21], Tong Ren[3], Yan Li[4], Yuanyuan Li[5], Lei Wu[6], Qiang Song[7], Mei Yang[2], Jian Zhang[9], Yuanyuan Liu[12], Shuliang Guo[14], Kun Yan[15], Xinghua Shen[16], Dan Lei[17], Yanli Zhang[20], Xiaofeng Yan[2]*, Liang Li[1]*, Shenjie Tang[1]*

**1** Beijing Chest Hospital, Beijing Tuberculosis and Thoracic Tumor Research Institute, Capital Medical University, Beijing, China, **2** Chongqing Public Health Medical Center, Chongqing, China, **3** Shenyang chest Hospital, Shenyang, China, **4** Guang Zhou Chest Hospital, Guangzhou, Guangdong, China, **5** Chest Hospital of Xinjiang, Urumqi, Xinjiang, China, **6** The Third People's Hospital of Kunming, Kunming City, Yunnan Province, China, **7** Shandong provincial Chest Hospital, Jinan, Shandong, China, **8** Jiangxi Chest (third people) Hospital, Nanchang City, Jiangxi Province, China, **9** Chang Chun Infectious Diseases Hospital, Changchun City, Jilin Province, China, **10** Taiyuan Fourth People's Hospital, Taiyuan City, Shanxi Province, China, **11** Fuzhou Pulmonary Hospital of Fujian, Fuzhou, China, **12** Tianjin Haihe Hospital, Tianjin City, China, **13** Third People's Hospital of Shenzhen, Shenzhen, China, **14** The First Affiliated Hospital of Chongqing Medical University, Chongqing, China, **15** Weifang NO.2 People's Hospital, Shandong Province, China, **16** The Fifth People's Hospital of Suzhou, Suzhou City, Jiangsu Province, China, **17** Affiliated Hospital of Zunyi Medical College, Zunyi, Guizhou, China, **18** The Fifth People's Hospital of Wuxi, Wuxi, China, **19** TB Hospital of Siping City, Siping City, Jilin Province, China, **20** Baoding Hospital for Infectious Disease, Baoding City, Hebei Province, China, **21** The First Affiliated of XiaMen University, Xiamen City, Fujian Province, China

☯ These authors contributed equally to this work.
\* tangsj1106@hotmail.com (ST); liliang@tb123.org (LL); 2429918342@qq.com (XY)

**Data Availability Statement:** All relevant data are within the manuscript and its Supporting Information files.

## Abstract

Tuberculosis (TB) remains a serious global public health problem in the present. TB also affects other sites (extrapulmonary tuberculosis, EPTB), and accounts for a significant proportion of tuberculosis cases worldwide. In order to comprehensively understand epidemiology of EBTB in China, and improve early diagnosis and treatment, we conducted a large-scale multi-center observational study to assess the demographic data and the prevalence of common EPTB inpatients, and further evaluate the prevalence of EPTB concurrent with Pulmonary tuberculosis (PTB) and the associations between multiple EPTB types and gender-age group in China. All consecutive age≥15yr inpatients with a confirmed diagnosis of EPTB during the period from January 2011 to December 2017 were included in the study. The descriptive statistical analysis included median and quartile measurements for continuous variables, and frequencies and proportions with 95% confidence intervals (CIs) for categorical variables. Multinomial logistic regression analysis was used to compare the

**Funding:** The research was conducted at the Key Project of Chinese National Programs (Grant No. 2015ZX10003001) to ST, and 'Beijing Municipal Administration of Hospitals' Ascent Plan (No. DFL20181601) to WK, Tongzhou District Science and Technology Committee [No. KJ2017CX054] to WK, and Tongzhou development Support Programme for High Level Talents (YHLD2019035) to WK. The funders had no role in study design, data collection and analysis, decision to publish, or preparation of the manuscript.

**Competing interests:** The authors have declared that no competing interests exist.

association of multiple EPTB types between age group and gender. The results showed that the proportion of 15–24 years and 25–34 years in EPTB inpatients were the most and the ratio of male: female was 1.51. Approximately 70% of EPTB inpatients were concurrent with PTB or other types of EPTB. The most common of EPTB was tuberculous pleurisy (50.15%), followed by bronchial tuberculosis (14.96%), tuberculous lymphadenitis of the neck (7.24%), tuberculous meningitis (7.23%), etc. It was found that many EPTB inpatients concurrent with PTB. The highest prevalence of EPTB concurrent with PTB was pharyngeal/laryngeal tuberculosis (91.31%), followed by bronchial tuberculosis (89.52%), tuberculosis of hilar lymph nodes (79.52%), tuberculosis of mediastinal lymph nodes (79.13%), intestinal tuberculosis (72.04%), tuberculous pleurisy (65.31%) and tuberculous meningitis (62.64%), etc. The results from EPTB concurrent with PTB suggested that females EPTB inpatients were less likely to be at higher risk of concurrent PTB (aOR = 0.819, 95% CI:0.803–0.835) after adjusted by age. As age increasing, the trend risk of concurrent PTB decreased (aOR = 0.994, 95%CI: 0.989–0.999) after adjusted by gender. Our study demonstrated that the common EPTB were tuberculous pleurisy, bronchial tuberculosis, tuberculous lymphadenitis of the neck, tuberculous meningitis, etc. A majority of patients with pharyngeal/laryngeal tuberculosis, bronchial tuberculosis, tuberculosis of hilar/mediastinal lymph nodes, intestinal tuberculosis, tuberculous pleurisy, tuberculous meningitis, etc. were concurrent with PTB. Female EPTB inpatients were less likely to be at higher risk of concurrent PTB, and as age increasing, the trend risk of concurrent PTB decreased. The clinicians should be alert to the presence of concurrent tuberculosis in EPTB, and all suspected cases of EPTB should be assessed for concomitant PTB to determine whether the case is infectious and to help for early diagnosis and treatment.

## Introduction

Tuberculosis (TB) remains a serious global public health problem in the present. According to the report of World Health Organization (WHO), the estimated global incidence of TB cases in 2018 was 10.0 million [1]. TB is a multi-systemic disease with a protean presentation. Pulmonary tuberculosis (PTB) is the most common clinical presentation of TB. TB also affects other sits (extrapulmonary tuberculosis, EPTB), such as pleura, lymph nodes, skeleton, meninges, etc [2,3]. In contrast to PTB, the research of EPTB is less concerned by public health institutions. This may be because most forms of EPTB do not contribute to the spread of tuberculosis [4,5], or sometimes patients with PTB and EPTB at the same time are classified as PTB cases [3]. In fact, EPTB accounts for a significant proportion of tuberculosis cases worldwide. Many studies have found a high proportion (20%-53%) of cases presenting with EPTB in all the cases of TB [5–8]. In clinical practice, concurrent other type of TB in EPTB patient is common [9,10]. Due to the atypical clinical symptoms, no characteristic imaging manifestation, difficulty in obtaining specimens, and low positive rate of etiology in EPTB, the diagnosis of EPTB is very difficult and easy to be misdiagnosed. Meanwhile, the presence of concurrent other types of TB in EPTB often complicates the diagnosis and clinical treatment, and may contribute to poor or unsatisfactory clinical outcomes in these patients. In order to comprehensively understand epidemiology of EBTB in China, and improve early diagnosis and treatment, we conducted a large-scale multi-center observational study to assess the demographic data and the prevalence of common EPTB inpatients, and further evaluate the prevalence of

EPTB concurrent with PTB and the associations between multiple EPTB types and gender-age group in China.

## Methods

In this study, EPTB patients were defined as those who have at least one EPTB disease site, including EPTB concurrent with PTB patients. According to WHO, EPTB cases were mainly categorized by the disease site [11]. The study was performed utilizing data of 21 hospitals from 15 provinces in China, most of which are specialized tuberculosis hospitals. All consecutive age≥15yr inpatients with a confirmed diagnosis of EPTB during the period from January 2011 to December 2017 were included in the study. Diagnosis of TB in these patients were undertaken utilizing WHO guidelines [12] and the clinical diagnosis standard for TB issued by the Chinese Medical Association [13]. In China, the diagnosis of tuberculosis generally adopts traditional and modern methods, relying on clinical symptoms and signs, together with the results of bacteriology tests, the tuberculin skin test (TST), X-ray /CT examination, pathological examination, T-SPOT.TB, Gene Xpert MTB/RIF assay, and the successful outcome of treatment with a course of anti-tuberculosis chemotherapy, etc. We referred to the patients' medical history record to collect medical and demographic information. Data collection and manuscript preparation were completed in accordance with the STROBE (Strengthening the Reporting of Observational Studies in Epidemiology) statement [14].

### Statistical analysis

Descriptive statistical analysis includes median and quartile measurements of continuous variables, and 95% confidence interval (CIS) frequency and proportion of categorical variables. Multinomial logistic regression analysis was used to compare the association of multiple EPTB types between age group and gender. Primary analyses were based on tests for trends, modeling the ordered age categories as linear terms. A $p$-value <0.05 was considered to be statistically significant. Odds ratios (ORs) with 95% confidence intervals (CIs) for age group and gender were calculated. All analyses were performed on the entire study population, as well as on six groups stratified by age (15–24 years, 25–34 years, 35–44years, 45–54years, 55–64years and≥65 years) and by gender. All data are sorted out in Microsoft Office Excel (Microsoft, Redmond, WA, USA) data table, and all analysis is conducted with SPSS software, version 13 (Chicago, USA).

### Ethics statement

This was a multi-center observational retrospective study. Given that the medical information of inpatients was recorded necessarily and anonymously by case history, and that our data analysis could not necessarily cause any breach of the privacy of, or present any undue personal risk to the participants in this study, the Ethics Committee of Beijing Chest Hospital, Capital Medical University, approved this study, with a waiver of informed consent from the patients involved.

## Results

### Patient characteristics

A total of 202,998 EPTB inpatients (age≥15 yr) from January 2011 to December 2017 were recruited at 21 hospitals from 15 provinces in China. And there were 416910 total TB patients. Demographic data for our patient cohort was shown in Table 1. The ratio of male: female was 1.51. The proportion of 15–24 years and 25–34 years in EPTB inpatients were the most.

**Table 1. The demographic data of EPTB inpatients.**

| Variables | N = 202998 | Proportion (%) |
|---|---|---|
| **Gender** | | |
| Female | 80860 | 39.83 |
| Male | 122138 | 60.17 |
| **Age group** | | |
| 15-24years | 42916 | 21.14 |
| 25-34years | 42569 | 20.97 |
| 35-44years | 28519 | 14.05 |
| 45-54years | 29671 | 14.62 |
| 55-64years | 26946 | 13.27 |
| ≥65years | 32377 | 15.95 |
| **EPTB types** | | |
| exclusively EPTB | 66249 | 32.64 |
| EPTB concurrent with PTB | 127005 | 62.56 |
| EPTB concurrent with other types of EPTB (without PTB) | 9744 | 4.80 |

Approximately 70% of EPTB inpatients were concurrent with PTB or other types of EPTB. The most common types of EPTB were EPTB concurrent with PTB (62.56%), followed by exclusively EPTB (32.64%), and EPTB concurrent with other types of EPTB (without PTB) (4.80%).

## The prevalence of most common of EPTB (0.4% or about ≥800 cases)

Our data analysis revealed that there were 364,694 TB lesions in 202,998 EPTB inpatients. On average each EPTB inpatient had 1.80 types of TB lesions. The most common of EPTB (0.4% or about ≥800cases) was tuberculous pleurisy (50.15%, 95%CI:49.93%-50.37%), followed by bronchial tuberculosis (14.96%, 95%CI: 14.80% -15.12%), tuberculous lymphadenitis of the neck (7.24%, 95%CI:7.13%-7.36%), tuberculous meningitis (7.23%, 95%CI:7.12%-7.35%), etc (Table 2).

## The prevalence of EPTB concurrent with PTB

It was found that many EPTB inpatients concurrent with PTB. The highest prevalence of EPTB concurrent with PTB was pharyngeal/laryngeal tuberculosis (91.31%), followed by bronchial tuberculosis (89.52%), tuberculosis of hilar lymph nodes (79.52%), tuberculosis of mediastinal lymph nodes (79.13%), intestinal tuberculosis (72.04%), tuberculous pleurisy (65.31%) and tuberculous meningitis (62.64%), etc. The EPTB(≥800 cases) concurrent with PTB (proportion≥30%) were shown in Table 3, sorted by proportion.

## The associations between multiple EPTB types and gender-age group

Table 4 illustrated the results of multinomial logistic regression analysis, with exclusively EPTB as the reference category. The results from EPTB concurrent with PTB suggested that females EPTB inpatients were less likely to be at higher risk of concurrent PTB (aOR = 0.819, 95%CI:0.803–0.835) after adjusted by age. As age increasing, the trend risk of concurrent PTB decreased (aOR = 0.994, 95%CI: 0.989–0.999) after adjusted by gender. When set ≥65years as the reference category, the risk of concurrent PTB firstly decreased then increased in EPTB inpatients.

**Table 2. The prevalence of most common of EPTB types (0.4% or ≥800 cases).**

| EPTB diagnosis | N | Proportion(95%CI) (%) |
|---|---|---|
| tuberculous pleurisy | 101808 | 50.15(49.93–50.37) |
| bronchial tuberculosis | 30367 | 14.96(14.80–15.12) |
| tuberculous lymphadenitis of the neck | 14706 | 7.24(7.13–7.36) |
| tuberculous meningitis | 14680 | 7.23(7.12–7.35) |
| tuberculous peritonitis | 9731 | 4.79(4.70–4.89) |
| tuberculous empyema | 7203 | 3.55(3.47–3.63) |
| lumbar vertebra tuberculosis | 7019 | 3.46(3.38–3.54) |
| tuberculous pericarditis | 5766 | 2.84(2.77–2.91) |
| thoracic vertebra tuberculosis | 5196 | 2.56(2.49–2.63) |
| tuberculous polyserositis | 4734 | 2.33(2.27–2.40) |
| intestinal tuberculosis | 4571 | 2.25(2.19–2.32) |
| chest wall tuberculosis | 4539 | 2.24(2.17–2.30) |
| tuberculosis of mediastinal lymph nodes | 3292 | 1.62(1.57–1.68) |
| renal tuberculosis | 2772 | 1.37(1.32–1.42) |
| pharyngeal and laryngeal tuberculosis | 2360 | 1.16(1.12–1.21) |
| pelvic tuberculosis | 1816 | 0.89(0.85–0.94) |
| tuberculosis of axillary lymph nodes | 1260 | 0.62(0.59–0.66) |
| knee joint tuberculosis | 1121 | 0.55(0.52–0.59) |
| tuberculosis of hilar lymph nodes | 1001 | 0.49(0.46–0.52) |
| pleural tuberculoma | 888 | 0.44(0.41–0.47) |
| hip joint tuberculosis | 878 | 0.43(0.40–0.46) |
| cutaneous tuberculosis | 849 | 0.42(0.39–0.45) |

Prevalence = N*100%/202998.

## Discussion

The clinical manifestations of tuberculosis (TB) are divided into pulmonary TB (PTB) and extrapulmonary TB (EPTB). PTB is much more frequent [3]. Extrapulmonary tuberculosis (EPTB) refers to the tuberculosis cases of extrapulmonary organs confirmed by bacteriology or clinically diagnosed, such as pleura, lymph nodes, abdomen, urogenital tract, skin, joints and bones, meninges, etc [11]. Global EPTB accounts for 13.37–53% of TB [6,7,15–20]. EPTB can coexist with PTB, or occur alone or other EBTB at the same time [9,10]. In fact, EPTB can present a variety of symptoms, which may be similar to the symptoms of other diseases, which poses a further challenge for diagnosis. In addition, getting the right samples to confirm EPTB is often seen as a challenge [21]. The ratio of PTB to EPTB varies with geographical, social, ethnic and economic parameters [22]. The composition ratio of EPTB varies greatly in different countries and continents, and the incidence of different types of EPTB varies greatly [9,20,23,24]. Our study showed that EPTB inpatients accounted for 48.69% of all TB patients. A prospective multicenter cohort study conducted in Thailand from May 2005 to September 2006 found that of 769 patients, only 461 (60%) were diagnosed with pulmonary TB, 78 (10%) with pulmonary and extrapulmonary tuberculosis, 223 (29%) with extrapulmonary tuberculosis, and extrapulmonary TB at more than one site in seven (1%) patients [25]. From January 2007 to December 2017, it was found that 73 patients (44.8%) had PTB, 71 patients (43.6%) had EPTB, 19 patients (11.7%) had PTB and EPTB at the same time in Turkey [26]. Of the 44,050 TB cases reported in Spain in 2007–2012, 31,508 (71.53%) were pulmonary tuberculosis and 12,542 (28.47%) were EPTB [27]. Almost one-fifth of TB cases in the United States are

**Table 3. The prevalence of EPTB (≥800 cases) concurrent with PTB (proportion≥ 30%).**

| id | EPTB | N1 | N2 | Proportion(95%CI) (%) |
|---|---|---|---|---|
| 1 | pharyngeal / laryngeal tuberculosis | 2360 | 2155 | 91.31(90.10–92.42) |
| 2 | bronchial tuberculosis | 30367 | 27185 | 89.52(89.17–89.86) |
| 3 | tuberculosis of hilar lymph nodes | 1001 | 796 | 79.52(76.89–81.98) |
| 4 | tuberculosis of mediastinal lymph nodes | 3292 | 2605 | 79.13(77.70–80.51) |
| 5 | intestinal tuberculosis | 4571 | 3293 | 72.04(70.72–73.34) |
| 6 | tuberculous pleurisy | 101808 | 66492 | 65.31(65.02–65.60) |
| 7 | tuberculosis of meningitis | 14680 | 9195 | 62.64(61.85–63.42) |
| 8 | tuberculous polyserositis | 4734 | 2811 | 59.38(57.96–60.78) |
| 9 | tuberculous pericarditis | 5766 | 3357 | 58.22(56.94–59.50) |
| 10 | tuberculous peritonitis | 9731 | 5574 | 57.28(56.29–58.27) |
| 11 | tuberculous empyema | 7203 | 3728 | 51.76(50.59–52.92) |
| 12 | chest wall tuberculosis | 4538 | 2195 | 48.37(46.91–49.83) |
| 13 | thoracic vertebra tuberculosis | 5196 | 2410 | 46.38(45.02–47.75) |
| 14 | pleural tuberculoma | 888 | 407 | 45.83(42.52–49.18) |
| 15 | tuberculous lymphadenitis of neck | 14706 | 6732 | 45.78(44.97–46.59) |
| 16 | cutaneous tuberculosis | 849 | 382 | 44.99(41.61–48.41) |
| 17 | tuberculosis of axillary lymph nodes | 1260 | 557 | 44.21(41.44–47.00) |
| 18 | renal tuberculosis | 2772 | 1193 | 43.04(41.18–44.91) |
| 19 | pelvic tuberculosis | 1816 | 777 | 42.79(40.50–45.10) |
| 20 | knee joint tuberculosis | 1121 | 454 | 40.50(37.61–43.44) |
| 21 | hip joint tuberculosis | 878 | 320 | 36.45(33.26–39.73) |
| 22 | lumbar vertebra tuberculosis | 7019 | 2515 | 35.83(34.71–36.97) |

N1: the number of EPTB; N2: the number of EPTB concurrent with PTB; Proportion = N2/N1.

extrapulmonary tuberculosis. In 253,299 cases, PTB accounted for 73.6% and EPTB 18.7% [6]. Information on some 626,093 cases of tuberculosis in Poland from 1974 to 2010 was collected. Out of 62,251 cases, extrapulmonary tuberculosis is the only form of disease (9.9% of all tuberculosis cases) [28]. To sum up, extrapulmonary tuberculosis is also a common type.

**Table 4. The associations between multiple EPTB types and gender-age group[a].**

| Types of EPTB | Variables | No. of concurrent PTB in EPTB (%) | aOR(95%CI) | P |
|---|---|---|---|---|
| **EPTB concurrent with PTB[b]** | Gender | | | |
| | Female | 48122(37.9) | 0.819(0.803–0.835) | <0.001 |
| | Male | 78883(62.1) | 1.0 | |
| | Age | | | |
| | 15–24 | 27094(21.3) | 0.986(0.955–1.017) | 0.374 |
| | 25–34 | 26806(21.1) | 0.992(0.961–1.024) | 0.627 |
| | 35–44 | 17193(13.5) | 0.876(0.847–0.907) | <0.001 |
| | 45–54 | 18154(14.3) | 0.886(0.856–0.917) | <0.001 |
| | 55–64 | 16835(13.3) | 0.921(0.890–0.954) | <0.001 |
| | ≥65 | 20923(16.5) | 1.0 | |
| | trend test | — | 0.994(0.989–0.999) | 0.030[c] |

[a]. The reference category was exclusively EPTB.

[b]. logit($P_{\text{EPTB concurrent with PTB}}/P_{\text{exclusively EPTB}}$)

[c]. the p value for trend

According to the data of this study, the demographic data of the patient cohort showed that male accounted for 60.17%, female 39.87%. The results are the same as those of Nepal, Turkey, Saudi Arabia and Poland [28–31]. A retrospective study of 102 patients with extrapulmonary tuberculosis from 2000 to 2004 was conducted at St. George's University Hospital in Pereira, Colombia. The average age of the patients was 31.6 years, and 62.7% of the patients were males [32]. A study from Turkey assessed the percentage and characteristics of cases of extrapulmonary tuberculosis (EPTB) in the industrial city of kokalai, Turkey. 636 cases were diagnosed with EPTB. There were 345 males (54.2%) and 291 females (45.8%). The average age of the patients was 22.5 +/- 17.1 years (1–86 years old); 41.4% of the patients were younger than 15 years old, 30.9% of the patients were between 20–39 years old [24]. Gender effect has a strong regulatory effect on the possible confounding factors, which indicates that a possible independent effect needs further study [33]. The gender effect in Yone et al., research has made a strong adjustment to the possible confusion, which indicates that a possible independent effect needs further study [33]. In general, it has been reported that the gender differences in TB manifestations are not clear [34]. Female patients with tuberculosis are more likely to have extrapulmonary manifestations than male patients, which may be related to the role of endocrine factors in the body [35]. Sreeramareddy et al., believe that in a high burden country like Nepal, women may be an independent risk factor for EPTB. TB control programs may target young people and women to find EPTB cases [29]. Jung et al., also found that women were independent predictors of EBTB in patients with active PTB (ratio 4.35, 95% CI: 1.78–10.63) [36]. The continued transmission of tuberculosis among the American born population is one of the causes of the difference in the incidence rate of gender specific diseases in San Francisco. Gender differences in incidence rate of tuberculosis may be due to differences in communication kinetics, rather than deviation from diagnosis or reporting [37]. Cellular immunity, hormones, iron metabolism, loss of income and financial barriers, as well as stigma may all be related to sex-specific differences [38–40], gender is just one of the factors [33]. Women are also the risk factors of high degree of stenosis. Women tend to be more active in EBTB type and multi-level EBTB.

Our study showed that the proportion of 15–24 years and 25–34 years in EPTB inpatients were the most. A Turkish study also showed that most cases of extrapulmonary tuberculosis occur in the 20-40-year age group [31]. EPTB was more common at younger ages (< 25 years) in Nepal [29]. The proportion of extrapulmonary tuberculosis in the population aged 0–19 in Poland is higher than that in other age groups [28]. Extrapulmonary tuberculosis is more common in Saudi Arabia at a young age (20–29) [19]. Sreeramareddy et al., also confirmed that youth may be an independent risk factor for EPTB in high burden countries [29]. Why the elderly are not prone to extrapulmonary tuberculosis may be related to the change of immune function in the elderly [41,42]. In the elderly, about 90% of TB cases are due to reactivation of primary infection. 30% to 50% of individuals may develop disease-free persistent infection. Some old people who have been infected with *Mycobacterium tuberculosis* before may eventually eliminate the living Mycobacterium tuberculosis and return to the tuberculin negative reaction state [43]. Generally, there is no consensus on the possible impact of aging on the occurrence of EPTB. It has been suggested that the failure of the immune system with age is related to EPTB, especially miliary and meningeal tuberculosis [44]. With the increase of age, the production of many cytokines is a common phenomenon. However, this increase was only evident in the culture stimulated by mitogen. In fact, there is no difference in the spontaneous production of all cytokines between young and old people. This finding not only shows that the cellular mechanisms for the production of these cytokines are best preserved in the cells of the elderly, but also that they can up regulate their production under appropriate stimulation [45]. This can happen in inflamed areas of the body, or in other damaged areas, where they

can make a significant contribution to the ongoing destructive process [46]. One of the physiological characteristics of immune aging may not be accidental, that is, the serum level of most immunoglobulins and subclasses increases [47], and the development of monoclonal antibodies is a significant pathological feature of the immune system in the elderly [48]. Peto et al., also found that children under 15 years were more likely to have EPTB than those over 15 years [6]. A Polish study also showed that the proportion of extrapulmonary tuberculosis in the population aged 0–19 was higher than that in other age groups [28]. This may be related to the special physiological characteristics and immune function of children and adolescents. In addition, during bacteremia or blood borne disseminated tuberculosis, which is also the cause of extrapulmonary tuberculosis, children and infants may develop the disease [49,50].

Our data analysis revealed that there were 364,694 TB lesions in 202,998 EPTB inpatients. On average each EPTB inpatient had 1.80 types of TB lesions. The most common of EPTB (0.4% or about ≥800cases) was tuberculous pleurisy (50.15%, 95%CI:49.93%-50.37%), followed by bronchial tuberculosis (14.96%, 95%CI: 14.80% -15.12%), tuberculous lymphadenitis of the neck (7.24%, 95%CI:7.13%-7.36%), tuberculous meningitis (7.23%, 95%CI:7.12%-7.35%), etc. Other studies have also found that pleural TB is the most common EPTB in sub Saharan Africa(63.2%, 144/228), Poland (36%, 214/599) and Romania (58%, 1606/2781) [21,28,33]. However, other studies found that lymph node tuberculosis is the most common sites of extrapulmonary tuberculosis in the Netherlands (39%, 1963 / 504239%), the United States (40%, 19107 / 47293), the United Kingdom (37%, 10358 / 27762), Saudi Arabia (44.6%, 170 / 381), Turkey (21%, 29 / 141) and Afghanistan (37.3%, 44 / 118) [6,7,23,51–53]. Some studies have also shown that the most common type of EPTB is bone and / or joint tuberculosis [54,55]. Of the 397 study patients in Turkey, 103 (25.9%) had EPTB and 294 (74.1%) had PTB. The two most common types of EPTB were genitourinary tuberculosis (27.2%) and meningeal tuberculosis (19.4%) [30]. This kind of different researches and different countries have different forms of common extrapulmonary tuberculosis, which may be related to the following factors: (1) in some countries, the immune function of the population is damaged due to vitamin D deficiency, dietary changes and restrictive social conditions, which leads to endogenous tuberculosis infection easily reactivated from the external parts of the lung, such as lymph nodes, pleura, bone tissue, etc., leading to EPTB in these parts High incidence [56]; (2) Genetic factors. NRAMP1 may play a more important role in the localization of Mycobacterium tuberculosis infection than at the beginning of infection, i.e., NRAMP1 could be a major gene governing human susceptibility to extra-pulmonary TB [57]. (3) The different genes of Mycobacterium tuberculosis strains may make patients susceptible to tuberculosis in some parts, such as lymph node tuberculosis [55]; (4) Bacillus Calmette-Guérin (BCG) immunization has different protective effects on various forms of tuberculosis, but it is not widely used in many countries [20,58]. Compared with countries without BCG vaccination, nationwide BCG vaccination in China may be related to different infection sites of extrapulmonary tuberculosis [20].

It was found that many EPTB inpatients were concurrent with PTB in our study. Laryngeal TB is an important form of tuberculosis. According to its pathogenesis, laryngeal TB can be divided into two types: primary, directly invading the larynx through bacteria; secondary, transmitted through the bronchi of advanced tuberculosis [59]. Laryngeal tuberculosis is often associated with pulmonary tuberculosis, but the patient also has tuberculosis of the larynx without a history of tuberculosis [60]. The incidence of the pulmonary tuberculosis rate of laryngeal tuberculosis was about 40.6% to 100% [61–63]. In our study, there were about 91.31% pharyngeal/laryngeal tuberculosis inpatients concurrent with PTB. In the study of Ling et al., the infection pattern of tuberculosis of larynx in the two groups seems to be different. Before 1990, the chest X-ray of 5 patients with larynx were all positive. After 1998, 8 out of 14

patients (57.1%) had normal chest radiographs and showed no signs of active lung disease [61]. Lim et al. found that the pulmonary tuberculosis rate of laryngeal tuberculosis was 80.0% (48/60) [62]. Clinical analysis of 22 patients with pathologically confirmed laryngeal tuberculosis was carried out retrospectively. The pulmonary tuberculosis rate of laryngeal tuberculosis was 59.0% (15/22) [63]. To sum up, it is assumed that the infection occurred via hematogenous spread from an unidentified source or was primarily laryngeal or spread from pulmonary TB [61,63,64]. Tracheobronchial tuberculosis (TBTB) is a kind of submucous tuberculosis, which mainly occurs in the bronchial mucosa and submucosa. TBTB can be misdiagnosed as bronchitis, bronchial asthma, bronchiectasis or pulmonary carcinoma. TBTB is present in 10%-40% among active tuberculosis patients, and the prevalence of TBTB is increasing in recent years [65]. Previous studies showed that the incidence of the pulmonary tuberculosis rate of bronchial tuberculosis was about 48% to 99.4% [66–70]. In our study, there were about 89.52% bronchial tuberculosis inpatients concurrent with PTB. Most of previous studies have reported that the prevalence of EBTB inpatients concurrent with PTB were higher, and respectively 99.4% [66], 92.1% [67], 91.9% [68] and 90.9% [69]. However, in the study of S Altin et al., only 48% (24/50) bronchial tuberculosis inpatients were concurrent with PTB [70]. The reason for different prevalence concurrent with PTB in EBTB is not clear. Some studies found that a number of patients with mediastinal tuberculous lymphadenitis were associated with lung parenchymal infection [71,72]. In our study, there were about 79.52% tuberculosis of hilar lymph nodes and 79.13% tuberculosis of mediastinal lymph nodes inpatients concurrent with PTB. Wojciech Rzechorzek et al., found that 50.0% (4/8) tuberculosis of hilar lymph nodes inpatients were concurrent with PTB [72]. However, another studies found that the incidence of the pulmonary tuberculosis in tuberculous lymphadenitis patients was lower (7.8% to 28.8%) [73–75]. Intestinal tuberculosis can result from the hematogenous spread of active tuberculosis or military tuberculosis, the adjacent spread of adjacent organs, the swallowing of active tuberculosis infected sputum or the intake of contaminated milk [76]. As a result, pulmonary involvement is not constant, ranging from 9.87% to 68.7% [76–80]. Our study showed that the prevalence of intestinal tuberculosis concurrent with PTB was 72.04%, consistent with the result of Kim et al. and Wang et al. [77,80]. Tuberculous pleurisy or pleural TB is a common manifestation of extrapulmonary TB [81–83]. In western countries, pleural tuberculosis makes up less than 1% of exudate and only 3–5% of tuberculosis patients [81,82]. In contrast, in developing countries like India, pleural TB accounts for 30–80% of all pleural effusion [84]. Previous studies reported the prevalence of associated parenchymal lesions ranged from 20 to 95.2% [85–89]. Our study found that the prevalence of PTB in the tuberculous pleurisy patients was 65.31%. CT scan can improve the accuracy of diagnosis by recording the related solid lesions and lymphadenopathy. 86% of the patients with tuberculous effusion showed parenchymal lesions on chest CT, and 37% showed radiation-induced tuberculosis [87]. Although tuberculous meningitis accounts for only 5–10% of extrapulmonary tuberculosis and 1% of all tuberculosis patients, it kills and maims more patients than any other form of tuberculosis [90–92]. In our study, there were about 62.64% tuberculous meningitis inpatients concurrent with PTB. Previous studies have reported that the prevalence of tuberculous meningitis inpatients concurrent with PTB were 29.4% (47/160) [93], 45.5% (95/209) [94], and 54% [95], respectively. A retrospective cohort study found that 58.8% (315/536) were associated with pulmonary tuberculosis, 12.3% were miliary tuberculosis [96]. Above studies demonstrated about 10%-100% of EPTB patients have concomitant pulmonary involvement [7,15,16,17,20,63]. Therefore, all suspected cases of EPTB should be assessed for concomitant PTB to determine whether the case is infectious and to assist with diagnosis.

Lastly, multinomial logistic regression analysis in our study showed that female EPTB inpatients were less likely to be at higher risk of concurrent PTB (aOR = 0.819, 95%CI:0.803–0.835)

after adjusted by age, and as age increasing, the trend risk of concurrent PTB decreased (aOR = 0.994, 95%CI: 0.989–0.999) after adjusted by gender. The similar results have not been reported in the previous study. The reasons remain to be investigated in the future.

There were several limitations to our study. Firstly, our study results may have been influenced by Berkson's bias. In our study, we selected all patients who were admitted to 21 hospitals with a diagnosis of PTB within a specified time period. There is a high likelihood that these hospitalized patients would have more concurrent tuberculosis in EPTB, relative to patients with EPTB not admitted to hospital. It is therefore likely that our study overestimates the prevalence of concurrent tuberculosis in EPTB patients. Secondly, most of the hospitals included in our study are specialized tuberculosis hospitals. Therefore, these findings may not represent the general EPTB patient population, and may not apply to settings elsewhere in China. Thirdly, many specialized TB hospitals in China do not admit or treat pediatric TB cases. Therefore our results did not study the prevalence of pediatric EPTB. Fourthly, we analyzed the distribution of EPTB by gender and age variables only, and did not collect data on other possible influencing factors, such as income, smoking, alcohol use, recreational drug use, housing arrangements, etc, which may affect the presence of concurrent tuberculosis in EPTB.

## Conclusion

Our study demonstrated that the common EPTB were tuberculous pleurisy, bronchial tuberculosis, tuberculous lymphadenitis of the neck, tuberculous meningitis, etc. A majority of patients with pharyngeal/laryngeal tuberculosis, bronchial tuberculosis, tuberculosis of hilar/mediastinal lymph nodes, intestinal tuberculosis, tuberculous pleurisy, tuberculous meningitis, etc. were concurrent with PTB. Female EPTB inpatients were less likely to be at higher risk of concurrent PTB, and as age increasing, the trend risk of concurrent PTB decreased. The clinicians should be alert to the presence of concurrent tuberculosis in EPTB, and all suspected cases of EPTB should be assessed for concomitant PTB to determine whether the case is infectious and to help for early diagnosis and treatment.

## Supporting information

**S1 Dataset.**
(XLSX)

## Acknowledgments

We acknowledge the contributions from Innovation Alliance on Tuberculosis Diagnosis and Treatment (Beijing), and the doctors, laboratory technicians and nursing staff at the 21 hospitals in China involved in this study.

## Author Contributions

**Data curation:** Wanli Kang, Shenjie Tang.

**Formal analysis:** Liang Li, Shenjie Tang.

**Investigation:** Song Yang, Hongyan Chen, Jianxiong Liu, Jinshan Ma, Mingwu Li, Jingmin Qin, Wei Shu, Peilan Zong, Yi Zhang, Yongkang Dong, Zhiyi Yang, Zaoxian Mei, Qunyi Deng, Pu Wang, Wenge Han, Meiying Wu, Ling Chen, Xinguo Zhao, Lei Tan, Fujian Li, Chao Zheng, Hongwei Liu, Xinjie Li, Ertai A, Yingrong Du, Fenglin Liu, Wenyu Cui, Quanhong Wang, Xiaohong Chen, Junfeng Han, Qingyao Xie, Yanmei Feng, Wenyu Liu, Peijun Tang, Jianyong Zhang, Jian Zheng, Dawei Chen, Xiangyang Yao, Tong Ren, Yan Li,

Yuanyuan Li, Lei Wu, Qiang Song, Mei Yang, Jian Zhang, Yuanyuan Liu, Shuliang Guo, Kun Yan, Xinghua Shen, Dan Lei, Yanli Zhang, Xiaofeng Yan.

**Methodology:** Shenjie Tang.

**Project administration:** Jian Du, Liang Li, Shenjie Tang.

**Resources:** Shenjie Tang.

**Software:** Shenjie Tang.

**Supervision:** Shenjie Tang.

**Validation:** Shenjie Tang.

**Writing – original draft:** Shenjie Tang.

**Writing – review & editing:** Jiajia Yu, Shenjie Tang.

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
