## [Decision Letter · Decision Letter 0]

20 Jul 2020

PONE-D-20-17556

The epidemiology of extrapulmonary tuberculosis in China: A large-scale multi-center observational study

PLOS ONE

Dear Dr. Tang,

Thank you for submitting your manuscript to PLOS ONE. After careful consideration, we feel that it has merit but does not fully meet PLOS ONE’s publication criteria as it currently stands. Therefore, we invite you to submit a revised version of the manuscript that addresses the points raised during the review process.

We look forward to receiving your revised manuscript.

Kind regards,

Hasnain Seyed Ehtesham

Academic Editor

PLOS ONE

Journal Requirements:

2. Thank you for indicating in the Methods section that your ethics board provided approval and that they waived the need for informed patient consent because the study was restrospective and used anonymized data. Please additionally add this information to the Ethics Statement.

4. Your ethics statement must appear in the Methods section of your manuscript. If your ethics statement is written in any section besides the Methods, please move it to the Methods section and delete it from any other section. Please also ensure that your ethics statement is included in your manuscript, as the ethics section of your online submission will not be published alongside your manuscript.

Reviewers' comments:

Reviewer's Responses to Questions

**Comments to the Author**

1. Is the manuscript technically sound, and do the data support the conclusions?

Reviewer #1: Yes

Reviewer #2: Yes

2. Has the statistical analysis been performed appropriately and rigorously? 

Reviewer #1: Yes

Reviewer #2: Yes

3. Have the authors made all data underlying the findings in their manuscript fully available?

Reviewer #1: Yes

Reviewer #2: Yes

4. Is the manuscript presented in an intelligible fashion and written in standard English?

Reviewer #1: Yes

Reviewer #2: Yes

5. Review Comments to the Author

Reviewer #1: The article is good as it presents a huge data collected from 21 hospitals in the country. In the limitations the authors have mentioned that they could not compare the demographic details as income, housing, smoking etc. They also mentioned that they could not procure much data about paediatric tuberculosis.

They have only mentioned number of tuberculosis cases and concurrent pulmonary tuberculosis cases.

However, the following points need to be mentioned:

1. BCG vaccination policy of China and the vaccination status of the patients.

2. Immune status of the patients with underlying illnesses.

3. The number of cases with previous TB infection.

4. Rate of cure and mortality.

The authors have presented huge amount of data but they haven't correlated the data with the above mentioned points and have not related to any demographic details, that is required to be done.

Reviewer #2: Comments:

As per title and objective, the article written by Kang et al. provides full illustration of epidemiology of EPTB in China. This observational study represented about 202,998 EPTB inpatients with collaboration of 21 hospitals from 15 provinces in China.

However, following suggestive comments will also construct this article stronger and beneficial for the readers:

Abstract:

• Word limits excided as per Journal guidelines. Should be modified while considering word limit.

• Abbreviations of PTB are missing.

Introduction:

• It is advised to add few points with recent references regarding diagnostic modalities and difficulties in diagnosing EP specimens as it is missing form introduction section.

Method:

• Author must specify about bronchial tuberculosis (14.96%), because this may consider under PTB.

Results:

• Risk of EPTB is related to the degree of exposure to the pathogen and host immune factors like HIV, diabetes, malignancy etc. It’s my opinion that if author have the related data then this could make article more encouraging to readers.

Discussion:

• Line no. 282: Mycobacterium tuberculosis , write in italic.

6. PLOS authors have the option to publish the peer review history of their article (what does this mean?). If published, this will include your full peer review and any attached files.

Reviewer #1: No

Reviewer #2: No

---

## [Author Response · Author response to Decision Letter 0]

28 Jul 2020

PONE-D-20-17556

The epidemiology of extrapulmonary tuberculosis in China: A large-scale multi-center observational study

PLOS ONE

Dear Academic Editor Hasnain Seyed Ehtesham,

On behalf of my co-authors, we thank you very much for giving us an opportunity to revise our manuscript. We have tried our best to revise our manuscript according to the comments. At the same time, the revised version has been uploaded.

We are pleased to answer the questions raised by the academic editor and reviewers and the manuscript has also been revised according to the comments.

Journal Requirements:

Answer: Thank you very much. Our manuscript meets PLOS ONE's style requirements. 

2. Thank you for indicating in the Methods section that your ethics board provided approval and that they waived the need for informed patient consent because the study was retrospective and used anonymized data. Please additionally add this information to the Ethics Statement. 

Answer: The information has been added to the ethics statement. 

Answer: I have an ORCID iD and that it is validated in Editorial Manager. 

4. Your ethics statement must appear in the Methods section of your manuscript. If your ethics statement is written in any section besides the Methods, please move it to the Methods section and delete it from any other section. Please also ensure that your ethics statement is included in your manuscript, as the ethics section of your online submission will not be published alongside your manuscript. 

Answer: The ethics statement has been added to the Methods of the manuscript. 

Reviewer's Responses to Questions

Reviewer #1: The article is good as it presents a huge data collected from 21 hospitals in the country. In the limitations the authors have mentioned that they could not compare the demographic details as income, housing, smoking etc. They also mentioned that they could not procure much data about paediatric tuberculosis.

They have only mentioned number of tuberculosis cases and concurrent pulmonary tuberculosis cases.

However, the following points need to be mentioned:

1. BCG vaccination policy of China and the vaccination status of the patients.

Answer: BCG vaccination policy of China: newborns are given BCG vaccination. BCG vaccination status of our group of patients was not investigated. 

2. Immune status of the patients with underlying illnesses.

Answer: Sorry, the immune status of this group of patients was not detected. 

3. The number of cases with previous TB infection.

Answer: Regrettably, previous TB infection in these patients was not included in our investigation. 

4. Rate of cure and mortality.

Answer: Rate of cure and mortality of this group of patients were not investigated in this survey. 

The authors have presented huge amount of data but they haven't correlated the data with the above mentioned points and have not related to any demographic details, that is required to be done.

Reviewer #2: Comments:

However, following suggestive comments will also construct this article stronger and beneficial for the readers:

Abstract:

• Word limits excided as per Journal guidelines. Should be modified while considering word limit.

Answer: According to the guidance of the magazine, the number of words in the abstract meets the requirements. 

• Abbreviations of PTB are missing.

Answer: It has been modified as required. 

Introduction:

• It is advised to add few points with recent references regarding diagnostic modalities and difficulties in diagnosing EP specimens as it is missing form introduction section. 

Answer: We have added the EPTB diagnostic difficulties in the introduction. 

Method:

• Author must specify about bronchial tuberculosis (14.96%), because this may consider under PTB. 

Answer: Tracheobronchial tuberculosis (TBTB) is a type of tuberculosis that occurs in the mucosa, submucosa, smooth muscle, cartilage, and membrane of the trachea and bronchial. Previously bronchial tuberculosis was classified as extrapulmonary tuberculosis in China. 

Results:

• Risk of EPTB is related to the degree of exposure to the pathogen and host immune factors like HIV, diabetes, malignancy etc. It’s my opinion that if author have the related data then this could make article more encouraging to readers.

Answer: Thank you very much for your suggestion. But, we don't have data on the degree of exposure to the pathogen and host immune factors in our survey. 

Discussion:

• Line no. 282: Mycobacterium tuberculosis, write in italic. 

Answer: In 282 lines of the manuscript, Mycobacterium Tuberculosis has been italicized.

Thank you very much for your attention and consideration. I am looking forward to hearing from you. 

Yours Sincerely,

Shenjie Tang

Beijing Chest Hospital, Capital Medical University, Beijing Tuberculosis and Thoracic Tumor Research Institute, Beijing, 101149, China

Email: tangsj1106@hotmail.com, tangsj1106@vip.sina.com

---

## [Editor Report · Decision Letter 1]

3 Aug 2020

The epidemiology of extrapulmonary tuberculosis in China: A large-scale multi-center observational study

PONE-D-20-17556R1

Dear Dr. Tang,

We’re pleased to inform you that your manuscript has been judged scientifically suitable for publication and will be formally accepted for publication once it meets all outstanding technical requirements.

Kind regards,

Hasnain Seyed Ehtesham

Academic Editor

PLOS ONE

Additional Editor Comments (optional):

I have gone through this revised manuscript and also the Author response to the comments of the Reviewers. Required information has been added to the Ethics Section and also ethics statements have been added to the Methods of the manuscript by the Authors. Missed Abbreviations of PTB is modified as required and Authors have also added the diagnostic modalities and difficulties in diagnosing EP specimens which was missing from the introduction section.The authors have satisfactorily addressed all the comments made by the reviewers and added all required information, and have revised the manuscript accordingly. I recommend this manuscript for publication.
---

## [Editor Report · Acceptance letter]

11 Aug 2020

PONE-D-20-17556R1 

The epidemiology of extrapulmonary tuberculosis in China: A large-scale multi-center observational study 

Dear Dr. Tang:

I'm pleased to inform you that your manuscript has been deemed suitable for publication in PLOS ONE. Congratulations! Your manuscript is now with our production department. 

Kind regards, 

on behalf of

Prof Hasnain Seyed Ehtesham 

Academic Editor

PLOS ONE